# Genome-Wide Association Study Reveals Host Factors Affecting Conjugation in *Escherichia coli*

**DOI:** 10.3390/microorganisms10030608

**Published:** 2022-03-12

**Authors:** Laetitia Van Wonterghem, Matteo De Chiara, Gianni Liti, Jonas Warringer, Anne Farewell, Natalie Verstraeten, Jan Michiels

**Affiliations:** 1Centre of Microbial and Plant Genetics, KU Leuven, 3001 Leuven, Belgium; laetitia.vanwonterghem@kuleuven.be (L.V.W.); natalie.verstraeten@kuleuven.be (N.V.); 2Center for Microbiology, Flanders Institute for Biotechnology, 3001 Leuven, Belgium; 3IRCAN, CNRS, INSERM, Université Côte d’Azur, 06107 Nice, France; matteo.dechiara@unice.fr (M.D.C.); gianni.liti@unice.fr (G.L.); 4Department of Chemistry and Molecular Biology, Centre for Antibiotic Resistance Research, University of Gothenburg, 405 30 Gothenburg, Sweden; jonas.warringer@cmb.gu.se (J.W.); anne.farewell@cmb.gu.se (A.F.)

**Keywords:** bacterial conjugation, horizontal gene transfer, plasmid, host factors, *Escherichia coli*, natural isolates, flow cytometry, genome-wide association study, antibiotic resistance, conjugation inhibitors

## Abstract

The emergence and dissemination of antibiotic resistance threaten the treatment of common bacterial infections. Resistance genes are often encoded on conjugative elements, which can be horizontally transferred to diverse bacteria. In order to delay conjugative transfer of resistance genes, more information is needed on the genetic determinants promoting conjugation. Here, we focus on which bacterial host factors in the donor assist transfer of conjugative plasmids. We introduced the broad-host-range plasmid pKJK10 into a diverse collection of 113 *Escherichia coli* strains and measured by flow cytometry how effectively each strain transfers its plasmid to a fixed *E. coli* recipient. Differences in conjugation efficiency of up to 2.7 and 3.8 orders of magnitude were observed after mating for 24 h and 48 h, respectively. These differences were linked to the underlying donor strain genetic variants in genome-wide association studies, thereby identifying candidate genes involved in conjugation. We confirmed the role of *fliF*, *fliK*, *kefB* and *ucpA* in the donor ability of conjugative elements by validating defects in the conjugation efficiency of the corresponding lab strain single-gene deletion mutants. Based on the known cellular functions of these genes, we suggest that the motility and the energy supply, the intracellular pH or salinity of the donor affect the efficiency of plasmid transfer. Overall, this work advances the search for targets for the development of conjugation inhibitors, which can be administered alongside antibiotics to more effectively treat bacterial infections.

## 1. Introduction

Antibiotic resistance poses a major challenge to human health. An estimation of 4.95 million deaths globally in 2019 was associated with antibiotic resistance, including 1.27 million deaths attributed directly to antibiotic resistance [1]. Among *Escherichia coli*, the cause of enteric/diarrhoeal disease, urinary tract infections and sepsis/meningitis, more than half of the isolates (57.1%) in the European Economic Area reported to EARS-Net in 2019 were resistant to at least one of the antimicrobial groups under surveillance, underscoring the wide spread of resistance [2,3]. Recently approved antibiotics and candidate drugs in the clinical pipeline should be shielded for as long as possible against the fast emergence and dissemination of resistance [4]. In this respect, conjugative DNA elements are especially problematic as they are notorious for capturing, retaining and then spreading antibiotic resistance genes [5,6]. Intra- and interspecies transfer within a patient or between a patient and the hospital environment can lead to local outbreaks and severely complicate treatment [7,8,9,10,11,12]. Therefore, conjugation inhibitors (COINs) present an interesting strategy to safeguard the efficacy of current and future antibiotics. These compounds are envisioned to be added during treatment alongside antibiotics to prevent the spread of resistance or could be used as pretreatment to increase the fraction of plasmid-free cells [13]. However, no COINs are currently in clinical use or in the advanced stages of development. The identification of potential targets for COINs, directed towards the donor, the recipient or the conjugative element, will be instrumental for the rational design of future therapies.

Theoretically, COINs can interfere with any of the steps of the conjugation process. In the donor cell, the relaxase and accessory proteins assemble at the origin of transfer (*oriT*) of the DNA element, forming the relaxosome. The relaxase nicks the DNA at the *nic* site and remains attached to the 5′ end. The coupling protein (T4CP) recruits the relaxosome to the type 4 secretion system (T4SS), which is anchored in the cell membrane of the host. Most likely, the adherence of the pilus to a recipient induces transfer of the relaxase–DNA complex to the recipient by the T4SS. This system is powered by ATPases. After the relaxase–DNA complex arrives in the recipient cell, the transferred DNA is re-circularised by the relaxase. Finally, the complementary strand of the transferred DNA is synthesized in both the donor and the recipient cell by rolling circle replication [14,15,16]. Potential COINs have previously been reported to impair conjugation by targeting the relaxase, the pilus or the traffic ATPase, which are encoded on the conjugative element itself. Unfortunately, these COINs only interfere with the transmission of a narrow spectrum of conjugative elements [17,18,19,20].

Horizontal transmission does not solely rely on genes carried by conjugative elements, but also on endogenous, chromosomally-encoded factors (further called host factors) in the donor or recipient cell. A *lux*-based screening of *E. coli* single-gene deletion mutants and transposon insertion mutants has demonstrated that inhibiting the lipopolysaccharide synthesis pathway in the recipient cell diminishes conjugation of the R388 plasmid [21]. Furthermore, a transposon sequencing (Tn-Seq) screen has revealed the role of phospholipid biosynthesis in the transfer of the integrative and conjugative element ICE*Bs1* in *Bacillus subtilis* [22,23]. Additionally, high-throughput monitoring of transconjugant growth, in which *E. coli* single-gene deletion mutants served as donors of the F plasmid, has revealed a role in the donor cell for DNA replication, chaperones or protein folding, and lipopolysaccharide core biosynthesis [24]. Finally, host factors have been described to interfere with the regulation of the transfer region of the conjugative elements [25]. For example, F plasmid transfer is stimulated by the anaerobic sensor ArcA and inhibited by the envelope stress sensor CpxA [26,27]. In contrast to the highly diverged plasmid-encoded proteins that constitute the core of the conjugation machinery, these bacterial host factors that have a more indirect role in conjugation are highly conserved and may therefore be promising targets for widely applicable COINs.

We here present a novel approach to disclose host factors involved in conjugation using flow cytometry to systematically measure the efficiency by which a diverse panel of *E. coli* strains transfer the broad-host-range IncP-1 plasmid pKJK10 to a fixed *E. coli* recipient strain. Subsequently, we tested for associations between the observed variation in conjugation efficiency and genotypic differences between donor strains in a hypothesis-free manner using genome-wide association studies (GWAS). Our analysis takes mutations, insertions, deletions, recombinations and the presence or absence of genes into account. Further validation in lab strain single-gene deletion mutants confirmed that the donor host genes *fliF*, *fliK*, *kefB* and *ucpA* positively affect the transfer of pKJK10.

## 2. Materials and Methods

### 2.1. Strains, Plasmids and Growth Conditions

The *E. coli* strains used in this study are listed in Appendix A [28,29,30,31]. Unless specified otherwise, strains were grown in LB medium at 37 °C. Liquid cultures were incubated in a shaking incubator at 200 rpm. When appropriate, additives were used at a final concentration of 10 µg/mL tetracycline (Tc), 40 µg/mL kanamycin (Km), 50 µg/mL trimethoprim (Tp), 35 µg/mL chloramphenicol (Cm), 50 µg/mL 5-aminolevulinic acid and 1 mM isopropyl β-D-1-thiogalactopyranoside (IPTG).

To construct strain BW25113 Δ*hemA::cat* (Cm^R^), a chloramphenicol resistance cassette was first amplified by PCR using pKD3 as template and primers with a 50-nucleotide tail homologous to the regions adjacent the *hemA* gene (ATGATGCAAGCAGACTAACCCTATCAACGTTGGTATTATTTCCCGCAGACcatatgaatatcctccttag and AGGCTTCATAGGCGTAAATGCACCCTGTAAAAAAAGAAAATGATGTACTGgtgtaggctggagctgcttc). The chromosomal *hemA* gene was subsequently replaced by the resistance cassette as previously described [32]. Plasmid pKJK10 was introduced into BW25113 Δ*hemA::cat* by chemical transformation.

The auxotrophic helper strain BW25113 Δ*hemA::cat* pKJK10 (Cm^R^, Tc^R^, Km^R^, Tp^R^) was used as donor to transfer pKJK10 to the natural strains by conjugation. For the latter, the natural strains were first streaked on agar plates containing Cm, Tc, Km or Tp to check for resistance. Next, BW25113 Δ*hemA::cat* pKJK10 and the natural strains were grown to stationary phase. The cultures were washed three times with PBS (pH 7.4) and the donor was concentrated 2-fold. The donor and the recipient were mixed in a 2:1 ratio and 100 µL was spotted on a polyvinylidene fluoride (PVDF) filter placed on top of an M9 minimal medium plate supplemented with 1% casamino acids. After overnight incubation at 30 °C, a small amount of inoculum was transferred by the streak plate method on M9 minimal medium plates supplemented with casamino acids, Tc, Km and Tp and incubated overnight at 37 °C to select for transconjugants. Inoculum of single-colonies was streaked again on M9 minimal medium plates with casamino acids, Tc, Km and Tp and incubated overnight at 37 °C. Due to the *hemA* deletion, the helper strain BW25113 Δ*hemA::cat* pKJK10 is defective in tetrapyrrole biosynthesis and is not able to grow on medium lacking 5-aminolevulinic acid [33]. We verified this by streaking the strain BW25113 Δ*hemA::cat* pKJK10 on M9 minimal medium with casamino acids and observed no growth. In addition, counter-selection of the donor strain was validated by streaking the cells on LB supplemented with 5-aminolevulinic acid and Cm, followed by overnight incubation at 37 °C. For each natural strain, the presence of pKJK10 was verified by PCR using primers targeting the relaxase gene encoded on pKJK10 (GGTGTGATCGAAACGGCA, CCTCAAGGGCAACACCGT). Eventually, BW25113 pKJK10 and the natural strains containing pKJK10 were used as donors for measuring conjugation efficiency.

Plasmid pSB1C3-mRFP1, kindly provided by Nicholas Coleman, was introduced into *E. coli* BW25113 by chemical transformation. BW25113 pSB1C3-mRFP1 (Cm^R^) was used as recipient for quantifying conjugation efficiency.

### 2.2. Quantifying Conjugation Efficiency

Donor and recipient cells were grown for 13 h in 5 mL LB at 37 °C. Cells were washed three times in PBS (pH 7.4) and resuspended in LB supplemented with IPTG, after which the OD (595 nm) was adjusted to 0.9. Next, 100 µL of donor and 100 µL of recipient cell suspension were mixed in an Eppendorf tube and incubated at 30 °C for 24 h and 48 h. We choose to incubate the samples at 30 °C since conjugation of pKJK10 or pKJK5 occurred in previous studies at 25 °C or 30 °C [34,35,36]. As controls, the donor and recipient were incubated separately in an Eppendorf tube. After 24 h, the Eppendorf tubes were vortexed, briefly opened to take 3 µL or 25 µL for dilution and measurement by flow cytometry or plating, respectively, and incubated again at 30 °C until the measurement at 48 h. Samples were kept on ice prior to measurement. All measured conjugation efficiencies are listed in Appendix A.

#### 2.2.1. Flow Cytometry

Measurements were carried out using a CytoFLEX S flow cytometer (Beckman Coulter). Particles of bacterial size were selected by applying a size threshold of FSC-A > 1305. Samples were diluted to obtain 30 to 300 recorded cell passage events per second, which ensured single-cell counting and signals exceeding the background noise. The controls were only diluted 100-fold because it was not required to distinguish different strains. Per sample, a minimum of 10,000 events were registered. In our set-up, donor cells harboured the conjugative plasmid pKJK10 encoding GFPmut3b, which rendered them green fluorescent [37]. Recipients were red fluorescent since they contained the non-mobilizable plasmid pSB1C3-mRFP1. Transconjugants contained pSB1C3-mRFP1 and pKJK10 and were therefore both green and red fluorescent. Fluorophores were excited by a 488 nm laser. Green fluorescence was measured through a FITC (525/40 BP) filter and red fluorescence was measured through a PE (585/42 BP) filter. The PE-A versus FITC-A plot was divided into four quadrants (with thresholds PE-A = 9.8 × 10^2^ and FITC-A = 2.7 × 10^4^) in order to discriminate between transconjugants, donors and recipients. The conjugation efficiency was expressed as the number of transconjugant events/the number of recipient events.

#### 2.2.2. Serial Dilutions and Plating

Serial dilutions of the conjugation mixture were plated on media selective for recipients (Cm) and transconjugants (Cm, Tc, Km, Tp). As control, the donor and the recipient were diluted separately, mixed in a 1:1 ratio and immediately plated out on media selective for transconjugants. All plates were incubated overnight at 37 °C. If the control plate contained more than 10 CFUs, the corresponding dilution from the conjugation mixture was not used since these transconjugants could have formed after plating. Conjugation efficiency was expressed as the number of transconjugant CFU/the number of recipient CFU.

### 2.3. Associating Conjugation Efficiency with Genetic Variants

The genome sequences of the 113 *E. coli* strains used in this study were determined for the TransPred project and can be retrieved with the SRA accession numbers given in Appendix A [38]. Genomes were assembled in SPAdes 3.13, with the k-mer length set to 33, 55, 67, 77, 87, 97, 107, 115, 125, and annotated in Prokka, specifying the input to the species *Escherichia coli* and using BW25113 (NCBI accession number: CP009273.1) as reference [39,40]. The phylogroup, sequence type and serotype were determined by the Web tools ClermonTyper, MLST 2.0 and SeroTypeFinder and listed in Appendix A [41,42,43]. Next, the following genetic variants were determined: single-nucleotide polymorphisms (SNPs), clusters of orthologous groups of proteins (COGs), k-mers and unitigs. VCF files describing SNPs were generated by mapping the paired-end reads against the BW25113 genome using snippy (with the option rename_cons on) [44]. The Gene Presence Absence Rtab file and the Core Gene Alignment fasta file were retrieved from Roary (minimum 95% identity for blastp, 100% of the isolates must contain the gene to be core) [45]. K-mers and unitigs were extracted using fsm-lite and unitig-caller, respectively [46,47].

The median of the measured conjugation efficiencies was log-transformed to reduce heteroscedasticity. In the case of ECOR-29, the median conjugation efficiency was set equal to the lower detection limit (1/10,000) because the logarithm of zero is undefined. The genetic variants were associated with the median log-transformed conjugation efficiencies by both a linear mixed model (LMM) and a fixed effects model in pyseer [48]. In both models, the genetic variants were included as fixed effect. To control for population structure, the linear mixed model uses a kinship matrix as random effect, while the fixed effects model uses an MDS decomposition of a distance matrix, in which 5 dimensions were retained, as fixed effect. Both the kinship matrix and the distance matrix were derived from the RAxML maximum-likelihood phylogenetic tree, which was calculated from the Core Gene Alignment [49]. A Bonferroni-corrected *p*-value threshold was calculated by dividing the significance level (α) by the number of unique variant patterns as the number of multiple tests. This determined which genetic variants reached genome-wide significance (α = 0.05) or suggestive significance (α = 1) [50,51].

### 2.4. Validating the Identified Genes by Single-Gene Knockout Strains

Based on the GWAS results, the plasmid pKJK10 was introduced by conjugation into a selection of single-gene knockout mutants from the KEIO collection (method described in ‘Strains, plasmids and growth conditions’). Next, the conjugation efficiency of the mutant strains containing pKJK10 was quantified by flow cytometry (method described in ‘Quantifying the conjugation efficiency’). As negative controls, we used BW25113 pKJK10 and BW25113 Δ*lacI::KmR* pKJK10, because *lacI* did not show an association with conjugation in the GWAS studies. Subsequently, the measured conjugation efficiencies were statistically analysed. First, normality was checked by plotting histograms and quantile-quantile (QQ) plots of the residual pool. Residuals were calculated as the difference between the observed log-transformed conjugation efficiency and the average log-transformed conjugation efficiency of the corresponding strain. Based on these plots, normality was rejected. Therefore, the non-parametric Kruskal–Wallis test was used to demonstrate for both 24 h and 48 h that the log-transformed conjugation efficiency differs between two or more strains (RStudio 4.1.0). Then, the post hoc non-parametric Dunn’s test was used to compare the mutant strains with the negative control BW25113 Δ*lacI::KmR* pKJK10 (GraphPad Prism 9.2.0). An additional analysis was performed with the non-parametric Conover test, which has greater statistical power compared to Dunn’s test, using a Bonferroni correction for multiple comparisons (RStudio 4.1.0, package DescTools). Adjusted *p*-values below 0.05 were considered significant.

### 2.5. Competition Experiment

We selected three *E. coli* strains (P06-59, P06-61 and P06-75) that were characterized by a low recipient fraction in the conjugation assay and two *E. coli* strains (P07-28 and BW25113) with a high recipient fraction and tested these strains with and without pKJK10 in competition with BW25113 pSB1C3-mRFP1, which was used as recipient in the conjugation assay. Cultures of the ten selected *E. coli* strains and the recipient strain BW25113 pSB1C3-mRFP1 were prepared, mixed and incubated as described for the conjugation assay. Additionally, all strains were incubated separately as control. After 0 h, 24 h and 48 h, samples were measured by flow cytometry as described for the conjugation assay. The different time points per strain were compared by a Kruskal–Wallis test (RStudio 4.1.0), a Dunn’s test (GraphPad Prism 9.2.0) and a Conover test with a Bonferroni correction for multiple comparisons (RStudio 4.1.0, package DescTools).

## 3. Results

### 3.1. Quantifying the Donor Abilities of Diverse E. coli Strains by Flow Cytometry

Our goal was to identify host factors in the donor cell that are involved in conjugation by associating the diversity in *E. coli* phenotypes with the diversity in genotypes. For this purpose, we measured the conjugation efficiency of a diverse panel of sequenced *E. coli* strains consisting of 1 lab strain, 59 strains isolated from different wild animals and 53 strains isolated from different human individuals (27 clinical isolates, 16 strains originating from healthy individuals and 10 strains of humans of unknown health). This collection comprises isolates from 79 different sequence types across the seven main phylogroups and the cryptic clade IV (29 in A, 32 in B1, 25 in B2, 2 in C, 17 in D, 2 in E, 5 in F and 1 in clade IV). The diversity of our collection is also reflected in the immunogenic structures since 49–50 different O-antigens and 34–35 different H-antigens were detected. For 22 strains, no O-antigen was found. We first transferred plasmid pKJK10 to each of the strains in the panel by conjugation. pKJK10 is a GFP-labelled derivative of pKJK5, a broad-host-range plasmid of incompatibility group P (IncP) isolated from soil in Copenhagen that confers resistance towards tetracycline, trimethoprim and to some extent spectinomycin [35,37,52]. We reasoned that since pKJK10 can be conjugated to very diverse bacteria [52], its transfer is more likely to depend on conserved host factors that could serve as valuable COIN targets.

Conjugation efficiencies are typically calculated from counting colony-forming units after plating. However, flow cytometry has previously been demonstrated to generate similar results in a less laborious manner [53]. We therefore optimised a flow cytometry assay capable of measuring the number of transconjugants, recipients and donors across a substantial number of matings (Figure 1). In our set-up, the conjugative plasmid pKJK10 confers green fluorescence to donor cells, while the non-mobilizable plasmid pSB1C3-mRFP1 renders recipients red fluorescent. Transconjugants harbour both the pSB1C3-mRFP1 and the pKJK10 plasmids and, consequently, display both green and red fluorescence. Separate cultures were used to set fluorescence thresholds for distinguishing donor, recipient and transconjugant cells from the background noise attributed to the solvent (Appendix A). Based on these thresholds, cells were called 24 h and 48 h after initiation of bacterial mating.

During the screening, diverse strains were tested as donors, while BW25113 pSB1C3-mRFP1 was always used as a recipient. Different expression levels of GFPmut3b were observed in the different donor strains, indicating that expression of GFPmut3b depends on the genetic background. However, a single genetic background was used as a fixed recipient strain and therefore guaranteed constant expression levels of mRFP1 in recipients, and GFPmut3b and mRFP1 in transconjugants. The conjugation efficiency was calculated as the number of transconjugants per recipient cell to ensure a correct comparison between matings. It is noteworthy that secondary transfer from transconjugants to plasmid-free recipients could be more efficient than primary transfer. In the case of the IncP-9 plasmid pWW0, the lag time before conjugation is shorter for transconjugants than for the donor, which could be attributed to temporary de-repression of plasmid genes in the transconjugant [54]. In addition, transfer of the IncP-α plasmid RP4 is more efficient to recipients with an identical strain background as the donor. It is possible that the plasmid is protected from restriction by shared restriction–modification systems in the donor cell and the recipient cell [55]. Nonetheless, the frequency of secondary transfers is inherently linked to the number of primary transfers from the original donor. Therefore, differences in conjugation efficiency are influenced only by the genetic variation among the primary donors.

As can be observed in Figure 2A,B, we found strong differences in conjugation efficiency across our *E. coli* panel. Indeed, the difference between the highest and the lowest median conjugation efficiency was 2.7 orders of magnitude after 24 h, and 3.8 orders of magnitude after 48 h. Overall, there is a strong positive correlation between the conjugation efficiencies measured after 48 h and those measured after 24 h (Figure 2C). A perfect correlation is lacking because the conjugation efficiency of certain strains stagnated after 24 h, while for other strains, the efficiency still increased between 24 h and 48 h. By measuring at two time points, both the fast-conjugating strains and the slow-conjugating strains are included in the GWAS analyses. We found that the recipient cell fraction varied between matings, dependent on the tested donor strain (Appendix A). For a selection of strains, we validated that the number of cells increased slightly in the absence of the recipient BW25113 pSB1C3-mRFP1, while the recipient fraction increased or decreased when the strains were incubated in the presence of BW25113 pSB1C3-mRFP1 (Appendix A, Appendix A). Therefore, the difference in recipient cell fraction between matings can likely be attributed to competition between the donor strain and recipient strain. However, this did not substantially affect the measured conjugation efficiency since only weak correlation was observed between the recipient fraction and the conjugation efficiency. Finally, we repeated the flow cytometric measurements for a subset of strains with varying conjugation efficiencies and in parallel measured their conjugation efficiency using plating on solid medium (Figure 2D). We found a strong positive correlation between the results obtained by the two methods, confirming that flow cytometric measurements reflect measurements through the classic plating well. Strain ECOR-04a was an exception since a high conjugation efficiency was measured by flow cytometry, while plating yielded a low efficiency. It is possible that conjugation and GFP expression was efficient, while the growth of the transconjugant was impaired.

Altogether, we have set up a simple and fast method to quantify conjugation efficiencies of a large collection of strains. Screening revealed that the strain background of the donor cell can have large effects on the conjugation efficiency. The measured conjugation efficiencies served as input for GWAS, as described in the next paragraph.

### 3.2. Associating Conjugation Efficiency with Genetic Variants

The availability of genome sequence information of our panel of *E. coli* strains allowed us to perform genome-wide association studies (GWAS). GWAS tests for an association between the phenotype of interest, in this case the conjugation efficiency, and genetic variants [56]. The genetic variants were single-nucleotide polymorphisms (SNPs), clusters of orthologous groups of proteins (COGs) or unitigs. SNPs are small substitutions, insertions and deletions in a DNA sequence compared to the reference genome [44]. A COG is a group of proteins with a high amino acid identity. COGs describe gene content and take non-reference sequences into account [45]. A unitig is a DNA sequence assembled from smaller fragments (also called k-mers), typically with a length of 31 nucleotides, and in which no competing choices are allowed. Unitigs (and k-mers) describe mutations, insertions, deletions and recombinations, and also include non-reference sequences [48,57].

We applied both a linear mixed model (LMM) and a fixed effects model in which the presence or absence of a genetic variant was included as a fixed effect. The population structure was taken into account as a random effect in the linear mixed model and as a fixed effect in the fixed effects model [58,59]. Both models estimated the effect of the genomic variant (β), which reflects the change in phenotype expected from carrying the genetic variant [56]. For every genetic variant, it was tested whether β differs significantly from 0. The resulting *p*-values are listed in Appendix A. In the mixed model, the -log (*p* values) better approximated a normal distribution compared to the fixed effects models, implying a higher reliability (Appendix A). Still, higher -log (*p* values) departed from the x = y reference line for COGs and unitigs. This can be attributed to conjugation being a truly polygenic trait and/or population stratification [56]. A Kruskal–Wallis test demonstrated that the phylogroup had a significant effect on conjugation efficiency after 24 h, but the effect was not significant after 48 h (*p* values 4.1 × 10^–3^ and 2.1 × 10^–1^, respectively) (Appendix A). Moreover, the clustering of strains with similar conjugation efficiencies can be observed in Appendix A. Significant and suggestive significant hits after SNP, COG and unitig association using the linear mixed model are summarized in Table 1, Table 2 and Table 3 Because rare variants can lead to spurious associations, only genetic variants with an allele frequency between 0.10 and 0.90 are listed as high confidence associations below (Appendix A). Overall, the genes associated with conjugation efficiency, differed between SNP, COG and unitig association. This likely reflects that conjugation is indeed a polygenic trait. Still, the genes identified through SNP and COG association also have a low *p*-value after unitig association, with the exception of the hypothetical protein (group_9935) in Table 2.

We concluded that genetic variants most intimately associated with conjugation occurred in genes involved in cell motility, anabolism and catabolism, transport of molecules or ions and modulation of the bacterial cell wall. In the next paragraph, a selection of genes will be validated for causality.

### 3.3. Validating Genes with Knockout Mutants

GWAS generated a list of chromosomally-encoded genes associated with conjugation. To verify if the identified genes are true hits, we selected corresponding single-gene knockout mutants from the *E. coli* Keio collection and measured their capacity to transfer the pKJK10 plasmid to a fixed recipient by flow cytometry (Figure 3) [28]. In addition to *fliF*, *tdh*, *gnd* and *yhhS* (Table 1 and Table 3), we tested genes identified by k-mer association (*yaiC*, *yecC*) and genes with an allele frequency > 0.90 (*ybeM*, *potA*, *ilvD*, *ydcN*, *eutK*), which all were identified using the linear mixed model. Gene *eutC* was tested instead of its downstream gene *eutK* since upstream deletion commonly affects downstream genes located in the same operon [60] and *eutC* was characterized by a low *p*-value in unitig association. Genes listed in Table 2 were not tested because deletion mutants of the complete gene are lacking in the Keio collection. We also tested genes that were called by the fixed effects model and performed well in the mixed effects model (*fliK*, *kefB*, *ucpA*, *yedQ*, *nlpC*, *yacH*, *otsA*, *uvrY*, *mdtB*, *yghJ*, *yedJ*, *uvrC*, *cheZ*, *iaaA*, *asmA*, *yecF*, *cysU*, *yncD*, *flhB*). As negative controls, we used the wild-type strain BW25113 and the Keio mutant BW25113 Δ*lacI::Km^R^*, which controlled for potential effects of the kanamycin cassette and showed no association to conjugation efficiency in the GWAS analysis (Appendix A). Whereas all genes selected for validation had low *p*-values in the GWAS analysis, not all were called as significant and therefore we expected some false hits.

A Dunn’s test demonstrated that the *fliF*, *fliK* and *kefB* knockouts differed significantly from the control BW25113 Δ*lacI::Km^R^* after 48 h. Additionally, a Conover’s test with Bonferroni correction for multiple comparisons was performed, which has more statistical power compared to a Dunn’s test. In this case, knockouts *fliF*, *fliK* and *kefB* differed significantly from BW25113 Δ*lacI::Km^R^* after 24 h, and knockouts *fliF*, *fliK, kefB* and *ucpA* differed significantly from BW25113 Δ*lacI::Km^R^* after 48 h. As expected, there was no significant difference between the negative control BW25113 Δ*lacI::Km^R^* and the wild-type BW25113.

In summary, we confirmed that genes *fliF* and *fliK* (both involved in the assembly of flagella), *kefB* (encoding a potassium transporter) and *ucpA* (encoding an oxidoreductase) positively affect conjugation. This demonstrates the capacity of GWAS to identify genes involved in conjugation.

## 4. Discussion

Our goal was to identify chromosomally-encoded genes affecting donor ability of the broad-host-range conjugative plasmid pKJK10. To reach this objective, we systematically measured the conjugation efficiency of diverse donors by flow cytometry and found a difference between the highest and the lowest median conjugation efficiency of 2.7 and 3.8 orders of magnitude after 24 and 48 h, respectively. In the next step, GWAS was applied to identify genes involved in conjugation by associating genetic variants with the observed conjugation efficiency. GWAS is an emerging technique originally used in humans and later successfully applied in bacteria to find determinants associated with resistance [61,62,63,64,65,66], virulence [51,67,68,69], host specificity [70], immunomodulation [71] and the host’s health [72]. We applied both a mixed effects model and a fixed effects model to test for association between genotype and phenotype and found that the mixed effects model was better suited to control *p*-values. However, genes identified using the mixed effects model did not exceed the genome-wide significance threshold and were merely suggestive significant. It is possible that the statistical power is impaired by the moderate size of our panel of *E. coli* strains and by the fact that conjugation is affected by multiple genes. Therefore, we decided to validate suggestive genes from the mixed effects model and several significant and suggestive genes from the fixed effects model. Conjugation assays with single-gene knockout mutants confirmed that *fliF*, *fliK*, *kefB* and *ucpA* positively affect conjugation in the lab strain BW25113. At this point, a potential role for the unconfirmed hits cannot be ruled out as these genes might affect conjugation when overexpressed, when point mutations are introduced, when tested in combination with other genes or when tested in a different strain background. Another possibility is that the observed associations of unconfirmed genes are attributed to shared ancestry rather than causality, a difficulty frequently encountered in bacterial GWAS [56].

Two of the genes that we report to be important for efficient conjugation are involved in the assembly of flagella: *fliF* and *fliK*. A flagellum is a filamentous organelle involved in bacterial motility and consists of a basal body as a motor, a filament as a helical screw and a hook as joint between the motor and filament [73]. The FliF proteins assemble into the MS ring in the basal body, which anchors the flagellum in the inner membrane [74]. Since the synthesis of flagella is costly, transcription of the operons is tightly regulated in three cascades [75,76]. FliK functions as a checkpoint controller by detecting when the hook has reached an appropriate length and triggers the switch from rod/hook-type to filament-type export [77]. It is hypothesized that FliK signals FlhB to switch substrate specificity, thereby triggering export of the antisigma factor FlgM and leaving the sigma factor FliA free to turn on class 3 genes [76,77]. In our validation experiments, the conjugation efficiency is impaired in *fliF* and *fliK* knockout mutants, indicating that bacterial motility or the presence of flagella stimulate conjugation. This is in line with earlier reports showing that changing the spatial structure of cells in a bacterial population increases plasmid invasion, most likely by increasing the number of contacts between potential donors and recipients or by increasing the probability that cells are in a location which is conducive for conjugation [78,79]. However, other studies have linked the carriage of conjugative plasmids with decreased motility [80,81,82,83,84,85]. In the case of the IncHI1 plasmid R27, the plasmid-encoded TrhR and TrhY regulators, which are required for induction of conjugation, downregulate flagella synthesis. TrhR/Y forms a regulatory circuit with the plasmid-encoded anti-activator HtdA. The absence of HtdA derepresses conjugation and causes a decrease in motility [80]. We hypothesize that while motility is important to increase the encounters between the donor and recipient in an appropriate location, it is diminished during the initiation of conjugation to increase the donor–recipient contact time and favour mating pair formation. This is supported by the finding that the conjugation efficiency of the *fliF* and *fliK* knockout mutants increases between 24 h and 48 h, knowing that cells were vortexed after 24 h. After 48 h, the mutants were still less efficient compared to the negative controls, indicating that the presence of flagella could assist mating pair formation. Interestingly, the interplay between motility and conjugation is observed for diverse plasmids (R27, pCAR1, pLS20, R1), which hints at a general mechanism [80,81,82,83,84].

To the best of our knowledge, this is the first study linking KefB and UcpA to conjugation. *kefB* encodes a potassium efflux system, which is inhibited by glutathione and activated by adducts formed by glutathione and an electrophile. Both the endogenously produced electrophile methylglyoxal and the externally added electrophile *N*-ethylmaleimide, in conjunction with glutathione, have been shown to activate KefB [86]. Glutathione plays a central role in the detoxification of electrophiles, thereby preventing damage to cellular macromolecules [87]. The activation of KefB results in a rapid loss of potassium and influx of protons, which in turn acidifies the cytoplasm. The lower intracellular pH protects cells against damage by electrophiles [86,88]. *ucpA* (upstream of *cysP*) encodes a cryptic oxidoreductase that reduces diacetyl to acetoin. Additionally, in the context of optimizing fermentation after pretreatment of lignocellulose, *ucpA* was found to increase resistance towards the inhibitory side product furfural by an unknown mechanism [89,90,91]. 

We envisage two ways in which *kefB* and *ucpA* could potentially promote efficient conjugation. First, their gene products might be involved in restoring balance in the glycolytic pathway. A glycolytic imbalance could be provoked by conjugation because the synthesis of the conjugative apparatus and the associated substrate transport is costly [16]. This cost is reflected in the need for tight regulation of transfer genes [25]. Methylglyoxal is mainly formed from the glycolytic intermediate dihydroxyacetone phosphate in a reaction catalysed by methylglyoxal synthase (MGS). The methylglyoxal bypass resolves an imbalance between the rate of carbon acquisition and the capacity of the lower segment of the glycolysis [92]. This prevents growth inhibition due to the accumulation of toxic sugar phosphates and increases the phosphate turnover in the cell, which can be advantageous when switching between carbon sources or growing in a phosphate-limited environment [92,93]. Acetoin is known to be formed as an overflow carbohydrate from the glycolysis and allows the regeneration of NAD^+^, which is required for the glycolysis to proceed [94,95]. Changes in central metabolic pathways have previously been associated with carriage of the conjugative plasmids pLS20 and pCAR1 [81,82]. In this respect, it is noteworthy that the gene for 6-phosphofructokinase 2 (*pfkB*), which catalyses the phosphorylation of fructose-6-phosphate during glycolysis, was significant in the fixed effects model with SNPs and unitigs.

Second, the pH decrease (from 7.6–8.0 to ~7.4) attributed to KefB and KefC activation might induce conjugation [87]. The pH dependency of conjugation has been observed for plasmids such as pQM1, pQM3, RP1 and pRK2073*::Tn5*, with optima for conjugation in the pH interval 6.0–7.5 [96,97,98]. UcpA, on the other hand, can contribute to restoring the intracellular pH by facilitating the conversion of the acid pyruvate to the neutral compound acetoin [99].

Additionally, KefB might stimulate conjugation by altering ion concentrations. It is known that moderate salt concentrations improve the transfer of different plasmids (e.g., pSLT, pESI, pUUH239.2), probably by decreasing electrostatic repulsion between cells [100,101,102]. Interestingly, the Na^+^/H^+^ antiporter NhaA and the potassium uptake protein KtrD have been found in earlier screenings to respectively affect the capacity of *E. coli* and *B. subtilis* cells to receive conjugative plasmids [21,22]. It could be beneficial for mating pair formation if the donor and/or the recipient is able to reduce electrostatic repulsion.

## 5. Conclusions

We have identified four chromosomally-encoded gene products that are involved in the *E. coli* donor ability of the broad-host-range conjugative plasmid pKJK10. In this respect, our work complements previous screenings searching for conserved targets for the development of COINs, which ultimately can be administered during treatment with antibiotics to halt the spread of antibiotic resistance [21,22,24]. Further work will elucidate the mechanisms whereby these gene products affect conjugation and explore whether their roles extend across a broader diversity of donors, recipients, plasmids and environmental conditions.

## Figures and Tables

**Figure 1 microorganisms-10-00608-f001:**
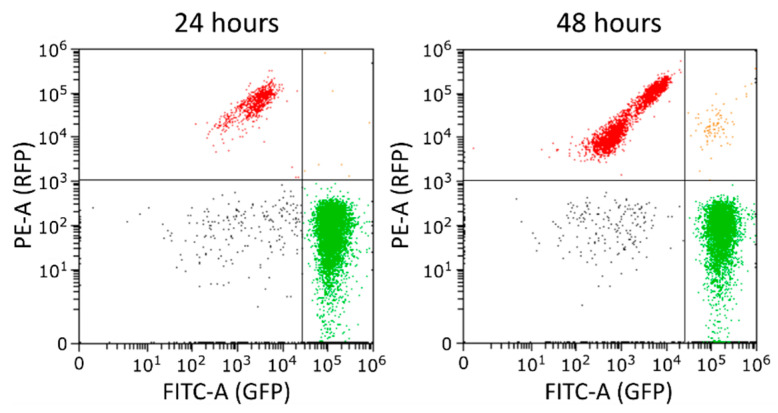
Quantification of conjugation efficiency by flow cytometry. The donor BW25113 pKJK10 and the recipient BW25113 pSB1C3-mRFP1 were mixed and measured after 24 h and 48 h. The PE-A versus FITC-A plot was divided into four quadrants (with thresholds PE-A = 9.8 × 10^2^ and FITC-A = 2.7 × 10^4^) in order to discriminate donors (green), recipients (red) and transconjugants (orange).

**Figure 2 microorganisms-10-00608-f002:**
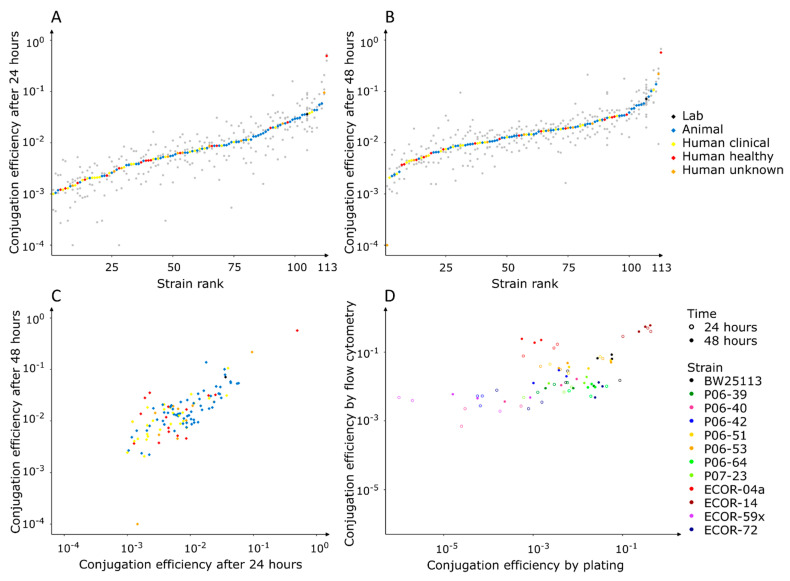
Screening of diverse *E. coli* strains as donors in conjugation. The conjugation efficiency was quantified after 24 h (**A**) and 48 h (**B**). Per strain, three biological replicas were tested (grey dots) and the median was calculated (coloured diamonds). (**C**) A positive correlation is observed between the median conjugation efficiency after 24 h and the median conjugation efficiency after 48 h (*r_Spearman_* = 0.73, *p* < 2.2 × 10^−16^). (**D**) A positive correlation is observed between measurement by flow cytometry and measurement by plating (*r_Spearman_* = 0.60, *p* = 3.53 × 10^−8^). In total, three biological replicas of twelve strains with varying conjugation efficiencies were quantified by flow cytometry and plating after 24 h and after 48 h.

**Figure 3 microorganisms-10-00608-f003:**
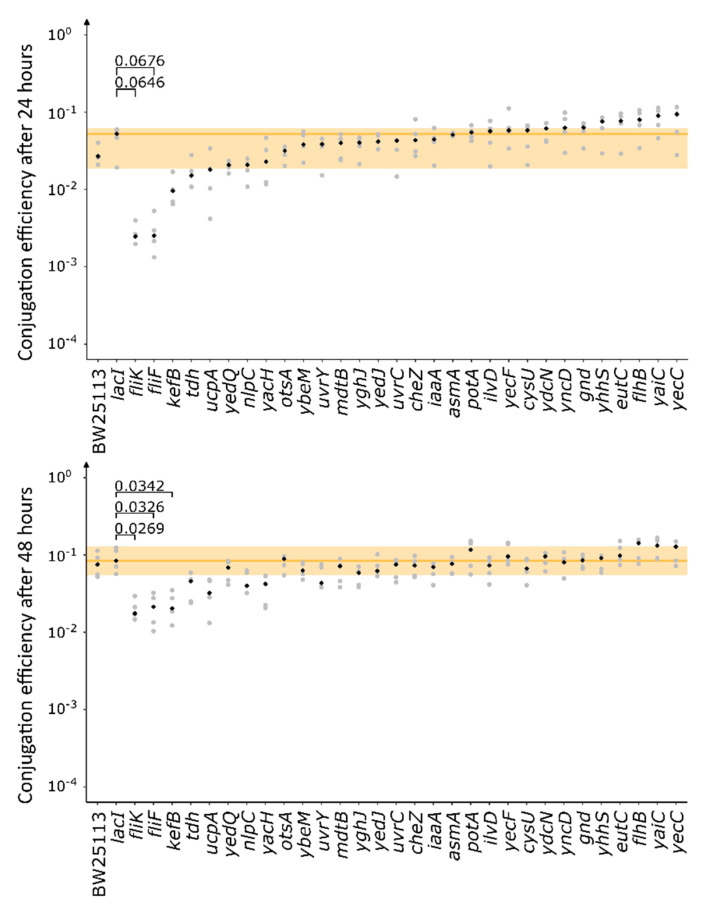
Validating genes identified by GWAS analysis. The conjugation efficiency of knockout mutants donating pKJK10 was quantified after 24 h and 48 h. Per strain, 5 biological replicas were tested (grey dots) and the median was calculated (black diamond). The range of the negative control BW25113 Δ*lacI::Km^R^* is coloured in yellow. The *p*-values following a Dunn’s test are indicated on the graphs. After 48 h, the knockout mutants of *fliF*, *fliK* and *kefB* differed significantly from the negative control BW25113 Δ*lacI::Km^R^* after 48 h. When a Conover’s test was performed, mutants of *fliF*, *fliK* and *kefB* differed significantly from BW25113 Δ*lacI::Km^R^* after 24 h with *p*-values 8.5 × 10^−4^, 9.4 × 10^−4^ and 1.0 × 10^−2^, respectively. After 48 h, mutants of *fliF*, *fliK*, *kefB* and *ucpA* differed significantly from BW25113 Δ*lacI::Km^R^* with *p*-values 5.3 × 10^−4^, 7.6 × 10^−4^, 8.4 × 10^−4^ and 1.4 × 10^−2^, respectively. The *p*-values of the Dunn’s test and the Conover’s test were adjusted for multiple comparisons and were considered significant below 0.05.

**Table 1 microorganisms-10-00608-t001:** Genes identified by SNP association with the log-transformed conjugation efficiency after 24 h and after 48 h using the linear mixed model. *p*-values lower than 3.47 × 10^–7^ are significant and lower than 6.93 × 10^−6^ are suggestive.

Time	Annotation		Variant Position: SNP	Effect Size β	Allele Frequency	*p*-Value SNPs	*p*-Value Unitigs
24 h	*tdh*	Threonine dehydrogenase	3783911: C→T	−0.61	0.63	4.34 × 10^−6^	5.19 × 10^−5^
48 h	*fliF*	Flagellar basal body MS ring and collar protein	2008002: C→T	0.70	0.11	5.56 × 10^−6^	2.67 × 10^−5^

**Table 2 microorganisms-10-00608-t002:** Genes identified by COG association with the log-transformed conjugation efficiency after 24 h and after 48 h using the linear mixed model. *p*-values lower than 3.86 × 10^−6^ are significant and lower than 7.72 × 10^−5^ are suggestive.

Time	Annotation		Number of Isolates	Effect Size β	Allele Frequency	*p*-Value COGs	*p*-Value Unitigs
24 h	group_9935 (*yncL* ^1^)	Hypothetical protein	13	1.00	0.12	2.37 × 10^−5^	3.08 × 10^−2^
48 h	*yedN*	Putative type III secreted effector	13	0.65	0.12	3.07 × 10^−5^	3.07 × 10^−5^

^1^ The sequence of *yncL* is embedded in the sequence of the hypothetical protein (Appendix A). In this case, the genome annotation tool Prokka chose a different open reading frame compared to the annotated sequence in BW25113.

**Table 3 microorganisms-10-00608-t003:** Genes identified by unitig association with the log-transformed conjugation efficiency after 24 h and after 48 h using the linear mixed model. *p*-values lower than 6.00 × 10^−8^ are significant and lower than 1.20 × 10^−6^ are suggestive. After 48 h, no gene with an allele frequency between 0.10 and 0.90 was significant or suggestive.

Time	Annotation		Variant	Effect Size β	Allele Frequency	*p*-Value
24 h	*gnd*	6-phosphogluconate dehydrogenase	CCAATATAGGTAACGCACGGTTCGCCATCTTCA	0.64	0.12	4.65 × 10^−^^7^
	*yhhS*	Putative transporter	GACCGCTCAAAAAGCAGCCGCATAAACCGAA	0.86	0.85	1.11 × 10^−^^6^

## Data Availability

Strains were sequenced for the TransPred project (http://transpred.ircan.org, accessed on 27 January 2022) These data are openly available in the Sequence Read Archive (SRA) under the accession numbers given in Appendix A.

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
