# Peer review of "Genome-Wide Association Study Reveals Host Factors Affecting Conjugation in Escherichia coli"

_microorganisms, 2022, doi:10.3390/microorganisms10030608_

Round 1

Reviewer 1 Report

In the manuscript “Genome-wide association study reveals host factors affecting conjugation in Escherichia coli”, Van Wonterghem and co-workers identify host factors in the donor cell that are involved in plasmid DNA conjugation. To do so, they use flow cytometry to measure the conjugation efficiency of the broad-host-range plasmid (IncP-1) pKJK10 carrying a green fluorescent reporter (GFPmut3b). One limitation of the experimental system presented is that the final conjugation efficiency measured results from donor-to-recipient primary transfer and transconjugant-to-recipient secondary transfer. However, this does not invalidate the relevance of the estimates since the extent of secondary transfer is intrinsically related to the frequency of primary transfers. In addition, this consideration is clearly presented by the authors.

It is worth noting that the significance of this study is reinforced by the fact that conjugation is tested from a diverse panel of 96 E. coli donors towards a single E. coli recipient strain. The donor panel includes one lab strain, 59 strains isolated from wild animals, and 53 strains isolated from human individuals (27 clinical isolates, 16 strains originating from healthy individuals and ten strains of humans of unknown health). It is an interesting finding per sethat this intra-species genetic diversity results in conjugation efficiency that varies up to 2.7 (after 24 hours) and 3.8 orders of magnitude (after 48 hours).

To reveal correlations between the donor strains’ genotypes diversity (including insertions, deletions, recombinations, and the presence or absence of genes) and the observed variation in conjugation efficiency, the authors use Genome-wide association studies (GWAS). For this specific analysis, the size of the panel (“only” 96 strains) is potentially too small to achieve satisfying statistical robustness. Nonetheless, the presented results convincingly emphasise the importance of several cellular pathways (including in cell motility, anabolism and catabolism, transport of molecules or ions, and modulation of the bacterial cell wall) for conjugation. These correlations are further validated using single-gene deletion mutants, among which single-gene knockout of fliF, fliK, kefB and upcA exhibit the expected effect on conjugation efficiencies.

The involvement of mobility through fliF and fliK effect mostly confirms previous reports that are well cited in the discussion section. By contrast, the finding that kefB (potassium efflux system) and upcA (cryptic oxidoreductase) influence conjugation is new. The authors discuss potential explanations for the effect of these genes in relation to their role in regulating the cell’s metabolic activity, intracellular pH or salinity, energy supply or reduction of electrostatic repulsion between donor and recipient. These interpretations are highly speculative yet appropriately discussed with suitable citations in the dedicated discussion section. Elucidating the particular role of these genes and pathways is clearly not in the scope of the article and should be the focus of future articles.

The work is sound, well written, and the interpretation are supported by the results presented. This manuscript validates the use of flow cytometry for conjugation efficiency estimates and GWAS to uncover genes involved in conjugation. Some results are confirmatory and not original (which is also validating the approach). Other results are new and open new questions, particularly regarding the interplay between the cell’s metabolism and its plasmid-donation proficiency. Overall, this work is a valuable addition to the corpus of previous publications investigating the role of the donor host genes on conjugation.

This reviewer has only to questions/comments.

Are the eutk, yedN and gene annotated as essential in the KEIO collection?

For instance, have the authors tried to construct this deletion mutant by standard methods such as lambda red recombination?

It is stated (l. 284 to l.288): “We found that the recipient cell fraction varied between matings, dependent on the tested donor strain (Figure S2). This can likely be attributed to competition between the donor strain and the recipient strain. However, this did not substantially affect the measured conjugation efficiency since only weak correlation was observed between the recipient fraction and the conjugation efficiency.” This reviewer thinks this is a crucial point that might be eluded a bit too quickly in the manuscript. The authors might be right that this effect is attributable to competition between the donor and the recipient strain in a given mating assay. This is easily testable by performing competition assays. Culture containing one plasmid-free donor and the recipient (2:1 ratio as for the conjugation assays) could be grown for 24h and 48h in the same conditions that the conjugation assays and the end-point ratio re-evaluated. Such an experiment, perhaps not decisive, would undoubtedly validate and strengthen the results and their interpretation.

Reviewer 2 Report

In this work, Wonterghem et al. apply GWAS to identify host factors affecting the conjugative transfer of an IncP plasmid. As, far as I know, this is the first work taking advantage of this approach to study this topic. Moreover, this is combined it with other state-of-the-art techniques, such as flow cytometry to measure conjugative efficiency. Overall, the work is therefore novel and innovative. The methodology and results are scientifically sound.

I have however a few comments that, I hope, can improve the work even more. My main comment is to include data regarding the strains, such as phylogroup, sequence type (ST) and serotype, which is easily done nowadays, for instance, with the tools at http://www.genomicepidemiology.org/. This information, could be added to Table 1, lines 238-242, and displayed similarly to Fig S2. It would help to discuss the importance of the phylogeny (indeed the authors comment on such effect in lines 331 and 424), as well as the potential role of LPS variation (discussed in line 85).

I would also request more details regarding the experiments. For example, in line 127-131: does M9 supplemented with casamino acids contain 5-aminolevulinic acid?; how were the cells transfered (were they resuspended, diluted and then seeded on plates)?. In line 452 it is mentioned that cells were vortexed at 24h but that was not clear from the methods section. Can the authors please clarify? 

In line 154,  why 30C? Are the cells in stationary phase between 24 are 48? Was the tube vortexed/opened at 24h allowing the culture to become aerobic again?

The results are honest and focus on suggested rather than significant genes. However, some sentences are confusing to me as they mention significant genes (e.g. line 376). Also, the sentence in line 418 seems to partially contradict the one line 414 by mentioning significant genes.

Although the sentence in line 273 is correct, it could deserve some discussion about the "high frequency transfer" (also know as "epidemic transfer") phenomenon observed in IncF plasmids where transconjugants transfer the plasmids more efficiently than the donors. As far as I know, this is not the case for IncP plasmids.

Some minor comments:
- The two paragraphs in lines 56-76 do not seem to be very important to the context of the results. The authors could consider shortening this information.
- In Table 1, the scientific species name is used for non-human hosts. For a matter of consistency, consider replacing human by Homo sapiens. 
- "donation" used to be a synonym of "mobilization". To avoid confusion I suggest to replace "donation" by "donor ability".
- In line 190: "33,55,67,77,87,97,107,115,125" - are these unformatted references?
- In line 372, yedJ should be in italics.
- Line 123 should read "strains" instead of "strain"
- In line 191, perhaps "reference" would read better than "trusted"?
